

# Large-scale gene flow in the barnacle *Jehlius cirratus* and contrasts with other broadly-distributed taxa along the Chilean coast

Baoying Guo[1] and John P. Wares[2]

[1] College of Marine Science and Technology, Zhejiang Ocean University, Zhoushan, Zhejiang, China
[2] Department of Genetics and Odum School of Ecology, University of Georgia, Athens, GA, USA

## ABSTRACT

We evaluate the population genetic structure of the intertidal barnacle *Jehlius cirratus* across a broad portion of its geographic distribution using data from the mitochondrial cytochrome oxidase I (COI) gene region. Despite sampling diversity from over 3,000 km of the linear range of this species, there is only slight regional structure indicated, with overall $\Phi_{CT}$ of 0.036 ($p < 0.001$) yet no support for isolation by distance. While these results suggest greater structure than previous studies of *J. cirratus* had indicated, the pattern of diversity is still far more subtle than in other similarly-distributed species with similar larval and life history traits. We compare these data and results with recent findings in four other intertidal species that have planktotrophic larvae. There are no clear patterns among these taxa that can be associated with intertidal depth or other known life history traits.

# INTRODUCTION

A persistent question in marine biogeography and population biology involves the interaction of species life history, geographic range, and trait or genealogical diversity within that range. In some cases, genealogical diversity or ''structure'' (*Wares, 2016*) *within* a species is informative of mechanisms that act to limit other species' distributional ranges (*Brante, Fernandez & Viard, 2012*; *Dawson, 2001*; *Riginos & Nachman, 2001*; *Wares, Gaines & Cunningham, 2001*). Of course, these studies often find that organisms with limited larval or juvenile dispersal have greater amounts of structure and less extensive ranges, but there are often exceptions (*Marko, 2004*). It is the variation among species, and the exceptions to the ''rules,'' that offer continued opportunity to understand marine diversity.

Early approaches to comparative phylogeography (*Dawson, 2001*; *Dawson et al., 2002*; *Hugall et al., 2002*; *Stuart-Fox et al., 2001*; *Sullivan, Arellano & Rogers, 2000*; *Wares, 2002*) focused primarily on regions of co-diversification of intraspecific lineages, e.g., the regions across which species were likely to exhibit structure. Subsequently, *Marko (2004)* noted that even when species had apparently identical life history and dispersal mechanisms, the distribution of a species across habitats (e.g., intertidal height) could influence their

Corresponding author
John P. Wares, jpwares@uga.edu

persistence in distinct glacial refugia. However, certainly to understand these associations more taxa should be compared, and *Kelly & Palumbi (2010)* made explicit comparisons of diversity and population divergence for 50 species along the Pacific coast of North America to suggest that species high in the intertidal were perhaps more likely to exhibit spatial genetic structure than those at lower depths. However, within taxa that are more closely related, e.g., among barnacles, this rule does not necessarily hold. Along the Pacific coast of North America, the high intertidal *Chthamalus dalli* exhibits no apparent population structure (*Wares & Castañeda, 2005*) relative to the mid-intertidal *Balanus glandula* (*Sotka et al., 2004*), while other barnacle species in this region also show effectively no structure (*Dawson et al., 2010*).

The particular spatial structure of the species represented in *Kelly & Palumbi (2010)* varies; however, there is often concordance of population structure—a non-random distribution of population discontinuities—among groups of species (*Pelc, Warner & Gaines, 2009*; *Small & Wares, 2010*) on this coast. Other regions that have been similarly explored—for example, the NW Atlantic coast—have fewer instances of strong population structure aside from regions that are also recognized as biogeographic transitions (*Altman et al., 2013*; *Díaz-Ferguson et al., 2009*) among more distinct groups of taxa. Another such example of this concordance of genetic diversity with biogeography was recently published by *Haye et al. (2014)*, looking at species with short-dispersing larval forms around the well-characterized biogeographic transition near 30°S latitude along the coast of Chile. Again, the structure of diversity within species was informative to the mechanisms—including shifts in upwelling intensity and nutrient availability (*Navarrete et al., 2005*)—that may limit the distribution of other taxa. As patterns of coastal upwelling are associated with phylogeographic structure in many regions and species (*Rocha-Olivares & Vetter, 1999*; *Zakas et al., 2009*), it merits exploration for how species respond to distinct oceanographic regimes along the Chilean coast.

Evaluating broad-scale diversity structure on the Chilean coast is of key interest as there are so many oceanographic and biogeographic comparisons to be made between this well-studied coastline and the well-studied Pacific coast of North America (*Navarrete, Broitman & Menge, 2008*). However, until recently there were few data available for species that spanned most of the length of the Chilean coastline. This scale is of interest because it spans *two* major biogeographic transitions—the region around 30°S noted above, as well as a notable biogeographic transition near 42°S (*Thiel et al., 2007*). While the divergence of taxa near 30°S is typically associated with shifts in upwelling and concomitant environmental transitions (*Ewers-Saucedo et al., 2016*; *Haye et al., 2014*), the biogeographic transition at 42°S is more likely driven by divergent current flow (*Ewers-Saucedo et al., 2016*). Temperature and salinity both exhibit significant transitions along this coastal region (*Acha et al., 2004*), and thus the dominant biogeographic boundary along the Chilean coast is at about 42°S (*Thiel et al., 2007*).

Some of the first such work at this spatial scale was done in the direct-developing gastropod *Acanthina monodon* (*Sanchez et al., 2011*) and another gastropod *Concholepas concholepas* (*Cardenas, Castilla & Viard, 2009*). In *Acanthina*, which has low dispersal potential among locations, strong concordance of intraspecific diversity with the 30°S

biogeographic boundary was found, but association with the 42° boundary was less clear. Nevertheless, statistically significant genetic structure and shifts in phenotypic diversity are associated with this region. The gastropod *Concholepas concholepas*, on the other hand, has high potential for pelagic larval dispersal, is similarly distributed along the coast of Chile, but exhibits no significant genetic structure at all (*Cardenas, Castilla & Viard, 2009*). These contrasts are wholly in line with predictions based on larval life history.

Recently, large data sets have become available for other commonly encountered taxa in the Chilean intertidal. Microsatellite data were analyzed in the mussel *Perumytilus purpuratus* (*Guiñez et al., 2016*), which both spawns gametes and has a long-lived planktotrophic larva, and this ecosystem engineer exhibited significant structure with two main lineages (separated at approximately 40°S) and isolation by distance within each lineage. Similarly, *Ewers-Saucedo et al. (2016)* explored genetic variation in the high intertidal barnacle *Notochthamalus scabrosus*, with nauplius larvae that have high pelagic larval dispersal potential, and found two primary lineages that mirror the dominant biogeographical pattern of Chile: in the northern Peruvian region only one lineage is found, while both are found in the Intermediate Area that represents the overlap of the Peruvian and Magellanic regions, and only the southern lineage is found south of 42°S. Another barnacle, the edible *picoroco* (*Austromegabalanus psittacus*) exhibits only slight structure along most of the Chilean coast (*Pappalardo et al., 2016*), but nevertheless the observed structure is statistically significant and seems to be associated with the northern (30°S) biogeographic transition.

To these data we add one more layer: *Zakas et al. (2009)* had explored mitochondrial sequence population structure in the high intertidal barnacle *Jehlius cirratus*, a species that is biologically and ecologically very similar to *Notochthamalus* but found slightly higher in the intertidal (*Lamb, Leslie & Shinen, 2014*; *Shinen & Navarrete, 2010*; *Shinen & Navarrete, 2014*). *Zakas et al. (2009)* found that unlike *Notochthamalus*, there was very little apparent genetic structure in *J. cirratus*. However, that analysis comprised only a small section of the Chilean coast from ∼28–34°S. Here we expand the sampling of *J. cirratus* to include diversity from ∼3,500 km of coastline, including most of the known distribution (*Häussermann & Försterra, 2009*). As chthamalid barnacles have a propensity to harbor cryptic genetic diversity (*Dando & Southward, 1981*; *Meyers, Pankey & Wares, 2013*; *Tsang et al., 2008*; *Wares et al., 2009*; *Zardus & Hadfield, 2005*), we specifically look for any phylogeographic structure that may add to our understanding of coastal biodiversity in Chile. We then more directly compare the whole-coast data described above for the ecological implications of the population structure identified within and among taxa.

## METHODS

Specimens of *J. cirratus* were collected from the intertidal in 2004–2013. Field permits were not required from the Subsecretaría de Pesca y Acuicultura for the specimens included in this paper, as they were not "shellfish resources." Sequences of cytochrome oxidase I ($n = 153$) from *Zakas et al. (2009)* were used in this study (Genbank GU126073–GU126226); additional sequences ($n = 187$) were generated from subsequent samples

**Table 1** Collection sites, number of individuals per sampling site (*n*) and summary statistics of genetic variability for *Jehlius cirratus*.

| Site (South Latitude) | Sampled | Haplotypes | Haplotype diversity | Nucleotide diversity ($\pi$) |
|---|---|---|---|---|
| Antofagasta/Arica (18.49°) | 31 | 27 | 0.978 ± 0.020 | 0.012 ± 0.009 |
| Huasco (28.46°) | 41 | 25 | 0.945 ± 0.022 | 0.009 ± 0.003 |
| Temblador (29.40°) | 21 | 16 | 0.948 ± 0.040 | 0.009 ± 0.006 |
| Guanaqueros (30.20°) | 24 | 18 | 0.942 ± 0.040 | 0.011 ± 0.006 |
| Punta Talca (30.95°) | 23 | 14 | 0.893 ± 0.052 | 0.008 ± 0.004 |
| Los Molles (32.25°) | 28 | 23 | 0.971 ± 0.024 | 0.011 ± 0.007 |
| Monte Mar (32.95°) | 28 | 24 | 0.987 ± 0.014 | 0.011 ± 0.006 |
| El Quisco (33.45°) | 29 | 25 | 0.988 ± 0.013 | 0.010 ± 0.006 |
| Las Cruces (33.49°) | 17 | 16 | 0.993 ± 0.023 | 0.012 ± 0.006 |
| Matanzas (33.95°) | 24 | 20 | 0.975 ± 0.024 | 0.011 ± 0.006 |
| Pichilemu (34.42°) | 32 | 24 | 0.958 ± 0.025 | 0.010 ± 0.008 |
| Niebla (39.85°) | 25 | 17 | 0.957 ± 0.024 | 0.014 ± 0.008 |
| Añihue (43.85°) | 8 | 7 | 0.964 ± 0.077 | 0.016 ± 0.009 |
| Isla Madre de Dios (50.42°) | 7 | 3 | 0.667 ± 0.160 | 0.009 ± 0.004 |

collected in 2011–2013 using PCR methods as in *Zakas et al. (2009)*. Samples were mostly collected in central Chile (Table 1), but this additional effort also added substantially to information from northern Chile and northern Patagonia.

After quality control and alignment of sequence data using CodonCode Aligner v6.0.2 (CodonCode Corporation), data were formatted for analysis using Arlequin v3.5.2.2. (*Excoffier, Laval & Schneider, 2005*) to identify population structure. Pairwise $\Phi_{ST}$ was calculated for all sites and compared to a matrix of pairwise geographic distance for signal of isolation by distance (*Wright, 1943*); this was done both with haplotypic data as well as nucleotide data under a K2P distance model. Additionally, an exact test of differentiation was calculated for all pairs of populations. Analysis of molecular variance (AMOVA) was performed to identify maximal structure along the coast as in *Dupanloup, Schneider & Excoffier (2002)* and *Zakas et al. (2009)* using an iterative approach for *K* contiguous spatial groups, increasing *K* if there were significant patterns of $\Phi_{SC}$ within the determined regional groups. Following the results of AMOVA, a haplotype network was generated using PopArt (http://popart.otago.ac.nz). Haplotypes were coded by sample location and by regions separated by the iterative AMOVA results that maximize $\Phi_{CT}$ to visually identify components of diversity associated with each regional group. Population diversity was also assessed at each sampled location; nucleotide diversity ($\pi$) and haplotype diversity (H) are estimated at each location using Arlequin.

## RESULTS

New sequences were archived in GenBank under accession numbers KX014910–KX015034. Site-specific diversity is presented in Table 1; pairwise values of $\Phi_{ST}$ are presented in Table 2. Only a single sequence was recovered from the northernmost collection site of Arica, so this sequence was included in the Antofagasta sample (results identical when excluded) for

Guo and Wares (2017), *PeerJ*, DOI 10.7717/peerj.2971

**Table 2  Pairwise $\Phi_{ST}$ values among sites (indicated as header) for mitochondrial COI sequence data in *Jehlius cirratus*.** Statistically significant ($p < 0.01$) comparisons are in bold. The sample from Antofagasta includes the single available sequence from Arica.

| Antofagasta | Huasco | Temblador | Guanaqueros | Punta Talca | Los Molles | Monte Mar | El Quisco | Las Cruces | Matanzas | Pichilemu | Niebla | Añihue |
|---|---|---|---|---|---|---|---|---|---|---|---|---|
| −0.10721 | | | | | | | | | | | | |
| −0.02397 | −0.10075 | | | | | | | | | | | |
| −0.06007 | 0.00344 | −0.09836 | | | | | | | | | | |
| −0.00797 | −0.07271 | 0.01272 | −0.01539 | | | | | | | | | |
| −0.01641 | −0.09486 | −0.01873 | −0.07157 | 0.00493 | | | | | | | | |
| −0.07084 | 0.01909 | −0.06296 | **0.05349** | −0.0808 | −0.03693 | | | | | | | |
| −0.17547 | −0.01582 | −0.18666 | 0.02576 | −0.1819 | −0.15953 | −0.03391 | | | | | | |
| −0.00509 | −0.06798 | 0.00201 | −0.02185 | −0.02005 | 0.01097 | −0.08597 | −0.16477 | | | | | |
| −0.07137 | 0.01015 | −0.05613 | **0.04841** | −0.0811 | −0.04482 | −0.0131 | −0.02592 | −0.07314 | | | | |
| **0.06509** | 0.01927 | **0.10959** | **0.10642** | 0.01976 | **0.085** | −0.01377 | −0.10077 | **0.04336** | −0.02223 | | | |
| −0.03313 | −0.0885 | 0.01678 | −0.04187 | −0.04029 | −0.02781 | −0.09641 | −0.21442 | −0.03887 | −0.10159 | −0.01699 | | |
| −0.01175 | 0.02556 | −0.00176 | 0.07232 | −0.03869 | 0.00933 | −0.03799 | −0.02988 | −0.04939 | 0.00464 | 0.02127 | −0.05271 | |
| −0.0777 | 0.01877 | −0.04544 | 0.08615 | −0.11043 | −0.08512 | 0.04286 | −0.00793 | −0.07119 | 0.03113 | −0.09806 | −0.13056 | 0.04426 |

**Table 3  Iterative AMOVA for $K = 2$ regions of sequence diversity.** Site is listed as dividing *that location and all sites to the north* from all locations to the south. The northernmost 2 sites (Arica, Antofagasta) were pooled for analysis as were the southernmost 2 sites (Añihue, Madre de Dios). Strongest values of $\Phi_{CT}$ (by magnitude and *p*-value) indicated in bold. Similar value of $\Phi_{CT}$ (0.0366, $p < 0.001$) is obtained with $K = 3$ and the regions separated as in Fig. 1.

| Site | $\Phi_{CT}$ | *p*-value |
|---|---|---|
| Huasco | 0.01406 | 0.16 |
| Temblador | 0.01977 | 0.11 |
| **Guanaqueros** | **0.03679** | **<0.001** |
| Punta Talca | 0.02623 | 0.03 |
| Los Molles | 0.03215 | <0.01 |
| Monte Mar | 0.02998 | 0.01 |
| El Quisco | 0.02896 | <0.01 |
| Las Cruces | 0.03463 | <0.01 |
| **Matanzas** | **0.03615** | **<0.005** |
| Pichilemu | 0.00076 | 0.55 |
| Niebla | 0.00635 | 0.64 |

statistical purposes. Values of $\Phi_{ST}$ were very low and in general not statistically significant (Table 2); the only exceptional locations were Guanaqueros (30°S) and Pichilemu (34°S), each of which tended to exhibit higher differentiation from a broader set of other locations. No population pairs were significantly different under an exact test. Testing these results for a pattern of genetic isolation by distance was not significant ($p$ 0.245).

Although negligible structure was exhibited along the Chilean coast in *J. cirratus* ($\Phi_{ST}$ of −0.019, $p$ ~1), there was statistical regional structure detectable with the increased power of sampling at that scale. Our implementation of spatial AMOVA (*Zakas et al., 2009*) recovered two contrasts for $K = 2$ regions in which $\Phi_{CT} > 0.035$ and $p < 0.01$, though similar results are found if the separation among regions is near to either of these locations (Table 3). These local maxima in $\Phi_{CT}$ separated Guanaqueros (30°S) and sites to the north from all locations to the south; and Pichilemu (44°S) and all sites to the south from all locations to the north. No significant $\Phi_{SC}$ was exhibited in these comparisons. If $K = 3$ groups are chosen using these same delineations, $\Phi_{CT}$ was comparable (0.03661, $p < 0.001$).

From these results, a haplotype network (minimum spanning tree) is presented in Fig. 1, showing "northern" diversity (from Guanaqueros northward), "southern" diversity (including Pichilemu and southward sites), and "central" diversity (locations in between), for visualization.

## DISCUSSION

As noted in *Zakas et al. (2009)* there is only slight population structure in *J. cirratus*. Previous efforts had also noted that using alternate statistics such as *Hudson (2000)* Snn also recovered no signal of structure or pattern of isolation by distance (*Wares, 2014*). Here, we identify statistically significant structure that is roughly associated with the 30°S biogeographic transition between the Peruvian and "Intermediate" zones, and there
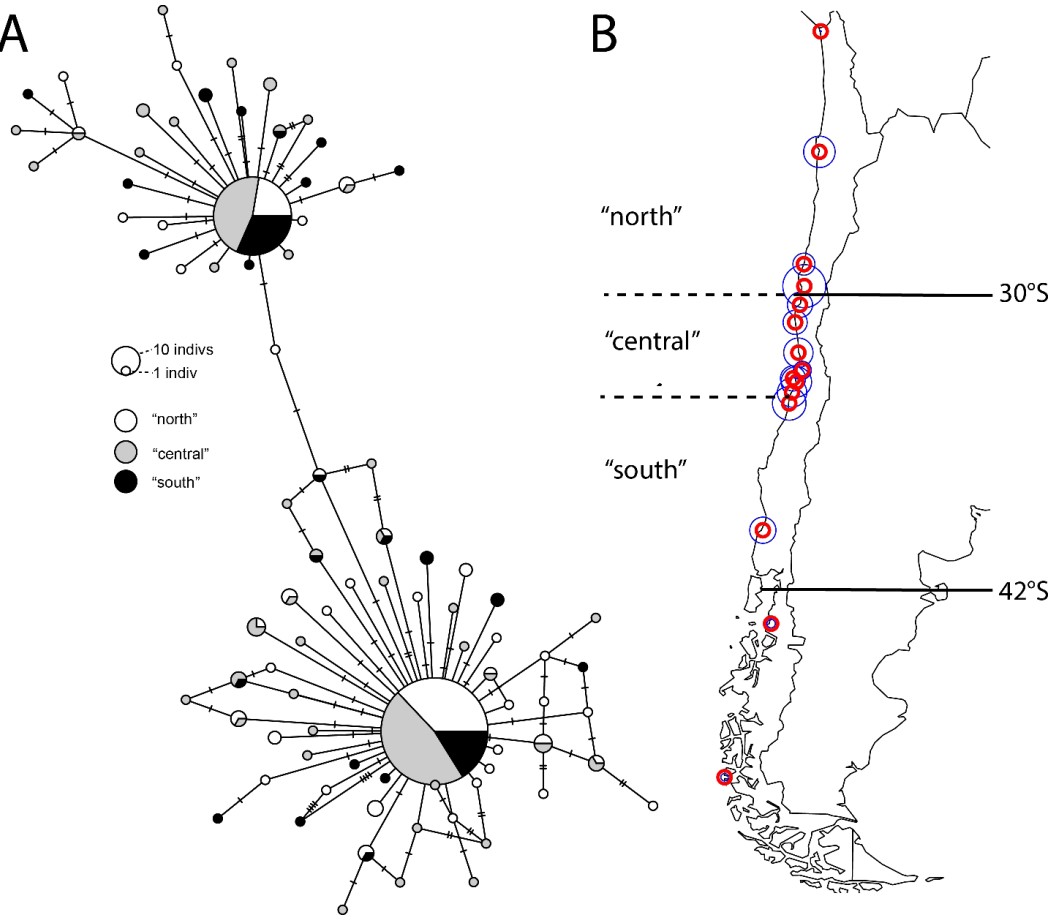

**Figure 1** **Patterns of regional diversity in *Jehlius cirratus* along the Chilean coast.** (A) Minimum-spanning tree of mitochondrial COI diversity in *J. cirratus*. Regional designations are generated from maximal $F_{CT}$ values along the coast. (B) The hypothesized transitions of species and genetic diversity noted from previous work (30°S, 42°S) and the regional separation of diversity supported by analyses of molecular variance in this study ("north," "central," and "south"). Red circles indicate sample locations along the coast; blue circles represent log-transformed sample size (see Table 1).

may also be structure further south—but not associated with the boundary at 42°S. Overall, the statistical significance indicated—given that pairwise statistical support was not consistent between permutational tests of $\Phi_{ST}$ and pairwise exact tests of population differentiation—suggests little actual spatial variation but sufficient sampling to identify the differential representation of regional samples in the 2 dominant haplotypes found (Fig. 1). Whether this is an instance of 'eurymixis' (*Dawson et al., 2011*), an instance where structure may exist but the power to detect it with available markers is insufficient to provide a consistent signal, is unclear. Nevertheless, the same methods have allowed the identification of phylogeographic structure in other species with similar distributions.

Excluding the direct developer *A. monodon* from further consideration, the studies reviewed earlier plus the current study include five intertidal species with high larval dispersal potential that are distributed and were analyzed along the length of the Chilean coast. Unfortunately, there is no clear pattern associated with intertidal depth; the species

with no or slight population genetic structure (*J. cirratus*, this study; *A. psittacus*, *Pappalardo et al., 2016*; *C. concholepas*, *Cardenas, Castilla & Viard, 2009*) are in the highest reaches of the intertidal (*J. cirratus*) as well as the low intertidal (*A. psittacus* and *C. concholepas*). The two species that exhibit significant structure, each with two primary lineages and evidence for isolation by distance within each lineage, are in the high-to-middle intertidal (*N. scabrosus*, *Ewers-Saucedo et al., 2016*, *P. purpuratus*, *Guiñez et al., 2016*).

Clearly a sample of only five taxa is insufficient for statistical consideration. However, what we can indicate is that all three barnacles (*A. psittacus*, *J. cirratus*, and *N. scabrosus*) have at least some signal associated with the 30–32° oceanographic transition in upwelling (*Lagos et al., 2005*; *Navarrete et al., 2005*); in contrast, the two molluscs, the mussel *P. purpuratus* and abalone *C. concholepas* do not. The association of genetic structure with the southern biogeographic boundary near 42°S (*Thiel et al., 2007*) is far more varied; other taxa with shorter distributional ranges that span this biogeographic transition, such as the mussel *Mytilus chilensis*, show little spatial structure at mitochondrial or other putatively neutral markers (L Besch & K Bockrath, 2015, unpublished data; *Araneda et al., 2016*) but can be distinguished among different coastal environments by outlier markers (*Araneda et al., 2016*) and expression profiling (*Núñez-Acuña et al., 2012*). *Ewers-Saucedo et al. (2016)* note that environmental transitions and current-mediated larval dispersal in this region, where trans-oceanic currents are separated as they reach the continental margin (*Acha et al., 2004*) are likely to transport regionally-differentiated diversity along a broad swath of this coastline. Thus, identifying concordant intraspecific diversity patterns among taxa may require a different analytical approach that is model-driven as in *Ewers-Saucedo et al. (2016)*.

There is an expanding interest in exploration of genetic diversity within and among regional populations of intertidal species along the coast of Chile (see *Haye et al., 2014* for a recent synthesis). Such data are being used to explore the underlying causes of biogeographic transition (*Cardenas, Castilla & Viard, 2009*; *Ewers-Saucedo et al., 2016*), to inform management and aquacultural concerns (*Haye & Munoz-Herrera, 2013*; *Núñez-Acuña et al., 2012*; *Pappalardo et al., 2016*), and better understand how the dynamics of a coastal ocean influence local diversity (*Aiken & Navarrete, 2014*; *Broitman et al., 2001*; *Hinojosa et al., 2006*; *Navarrete et al., 2005*). For example, even with variation among the data and taxa evaluated here, there is a concordance between the genetic transitions exhibited in these taxa and regions of strong upwelling along coastal Chile (*Navarrete et al., 2005*).

What remains unsatisfying is our ability to predict—based on what we know of life history, ecology, and other parameters of a given taxon—which species are likely to exhibit structure across a certain region, or why some species are able to spread across boundaries that others cannot (*Dawson, 2014*). *Haydon, Crother & Pianka (1994)* first noted the problem of both stochastic and deterministic contributions to biogeography and overall population structure. Certainly some 'significant' phylogeographic structure may simply represent the interaction of genealogical processes and modest limitations on gene flow (*Irwin, 2002*). However, the most direct contrast of the taxa included here involves the barnacles *N. scabrosus* and *J. cirratus*, which are ecologically nearly indistinguishable (*Lamb, Leslie & Shinen, 2014*; *Shinen & Navarrete, 2010*; *Shinen & Navarrete, 2014*) with

little known distinction in larval life history. In fact, though *N. scabrosus* exhibits significant phylogeographic structure (*Ewers-Saucedo et al., 2016*), the larvae of *N. scabrosus* appear to require longer times in the plankton and longer times for cyprid metamorphosis than *J. cirratus* (*Venegas et al., 2000*). Whether the cause for this contrast in population structure across a large geographic range is ecological, physiological, or simply chance remains unclear.

## ACKNOWLEDGEMENTS

The author would like to thank Sergio Navarrete, John Binford, Christina Zakas, Leah Besch, Jenna Shinen, Arnaldo Vilaxa Olcay, Daniel Saucedo, Ulo Pörschmann, and Christine Ewers-Saucedo for assistance in collecting specimens and sequence data. Karen Bobier, Paula Pappalardo, Katie Bockrath, Leah Besch, and Bud Freeman assisted with preparation of the manuscript.

### Funding

Funding for this project is from the National Science Foundation, Biological Oceanography panel (NSF-OCE-1029526). The funders had no role in study design, data collection and analysis, decision to publish, or preparation of the manuscript.

### Grant Disclosures

The following grant information was disclosed by the authors:
National Science Foundation, Biological Oceanography: NSF-OCE-1029526.

### Competing Interests

The authors declare there are no competing interests.

### Author Contributions

- Baoying Guo performed the experiments, analyzed the data, prepared figures and/or tables, reviewed drafts of the paper.
- John P. Wares conceived and designed the experiments, performed the experiments, analyzed the data, contributed reagents/materials/analysis tools, wrote the paper, prepared figures and/or tables, reviewed drafts of the paper.

### Field Study Permissions

The following information was supplied relating to field study approvals (i.e., approving body and any reference numbers):

Field permits were not required for the specimens included in this paper, as the Subsecretaría de Pesca y Acuicultura in Chile determined that barnacles were not "shellfish resources."

# PeerJ

## DNA Deposition

The following information was supplied regarding the deposition of DNA sequences:

GenBank accession numbers are reported for all sequences in the study. New sequences are archived in Genbank under accession numbers KX014910–KX015034.

## Data Availability

New sequences are archived in Genbank under accession numbers KX014910–KX015034.

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
