# Peer review of "Large-scale gene flow in the barnacle Jehlius cirratus and contrasts with other broadly-distributed taxa along the Chilean coast"

_PeerJ, doi:10.7717/peerj.2971_

## Round 0.1 · original submission · Major Revisions

I have heard back from two reviewers, both of whom were generally positive about your work. Both did, however, also offer many constructive comments regarding improvements that could be made to your manuscript. These changes will take some work and time, and therefore my decision is 'Major Revisions'.

Reviewer 1 ·

Basic reporting

In their current study, the authors analyse the genetic diversity of the species Jehlius cirratus using a COI dataset… The sampling is very comprehensive, data are analysed properly, and the paper reads well.

I understand that the authors are specifically interested in testing geographical barriers to dispersal that have been reported for other species. For that purpose they use an Iterative AMOVA approach, which succeeds in recovering 2 main genetic discontinuities along the Chilean coast. Very well! Yet alternative hypotheses are not tested and so it does feel a little bit like the authors are imposing their a priori hypothesis to the reader. But models will always fit a dataset (and not only in genetics)… The question is to know if that model with its additional parameters has a better explanatory power for a given dataset compared to a simpler model. Of course the risk of overfitting is very limited in the iterative AMOVA used here, since the only parameter optimise is the structure of the population itself. My concern is elsewhere: how this model compares to the two alternative hypotheses that are the island model (unlimited gene flow, H0) and a model of isolation by distance (Mantel’s model, H0’).
I am surprised the software you used does not provide at least a statistic for the island model, have you checked?
I believe it would have been fairly easy to formally test that in R (I think Ade4 implements AMOVA, see also packages Ape and Pegas)… Implement your model in that AMOVA, implement a Mantel test (with lm() function), implement the island model (AMOVA with no structure) and compare the results using a simple ANOVA. Unless you know another way which could really allow testing your model.
That is my main concern and I would like it to be somehow addressed in the review… That come next are more minor comments.


Site names are not consistent across the 3 tables… For instance the “Valdivia” that appears in table 3 appears nowhere else. Is it actually the “Niebla” of tables 1 and 2 (perhaps some unwanted autocorrect?). It is said in the caption of the table 3 that Arica was pooled with Antofagasta and Añihue was pooled with Madre de Dios… Does it apply to table 2 as well?

Figure 1: would it possible to replace those ugly circles by pie charts showing the haplotypic diversity (with size proportional to sample size if possible)?

l.130: (Dupanloup et al. 2002) -> Dupanloup et al. 2002
l.257: italicise Concholepas concholepas
l.274: italicise Perumytilus purpuratus
l.281: italicise Pyura chilensis
l.315: italicise Austromegabalanus Psittacus; Psittacus -> psittacus
l.336: italicise Chthamalus malayensis
l.345: italicise Jehlius cirratus

Experimental design

No Comments

Validity of the findings

No Comments

Additional comments

No Comments

Reviewer 2 ·

Basic reporting

This manuscript examine the population genetics of the barnacle Jehlius along broad latitudinal gradient in Chilean coastline. Results revealed two genetically distinct populations. The results are worth publishing but the introduction, discussion and figures need revisions and I can review this MS again.

Experimental design

Good and clear.

Validity of the findings

Finding are important.

Additional comments

Review of Guo.

This manuscript examine the population genetics of the barnacle Jehlius along broad latitudinal gradient in Chilean coastline. Results revealed two genetically distinct populations. The results are worth publishing but the introduction, discussion and figures need revisions as below:
Introduction – the authors stated “marine biogeography and population biology involves the interaction of species life history, geographic range, and trait or genealogical diversity within that range.” This is not totally correct as the biogeography and population biology also interact with the geological history (e.g. sea level rises and glacial refugia) and physical factors (wind and currents) that affect the gene flow of the species. This aspect has not been stated in the introduction. I would suggest the introduction should have brief mention what are factors to interact with population ecology of intertidal species in West Pacific and Pacific Coast of N and S America, before entering deeper into the situation of Chile.

In NW Pacific, the presences of marginal sea can create glacial refugia to affect gene flow of populations (see Magg’s models). There are many examples in barnacles in NW Pacific (Chthamalus malayensis, C. moro and Tetraclita) that have their population genetics interacted with geological factors and oceanographic currents. Although this is not always the case in the Pacific coast of South America. This should be mentioned in the introduction. In S. America, Galapagos Islands above the northern water of Chile is also a glacial refugia. The concept of glacial regufia should be stated in the introduction. There are no mentioning of what is the concept of isolation by distance in the introduction as well.

The introduction had not mention adequate examples of barnacles in Pacific coast of N. America. For example, Ford et al 1993 - Population structure of the pink barnacle, Tetraclita squamosa rubescens, along the California coast. Dawson et al. Population genetic analysis of a recent range expansion: mechanisms regulating the poleward range limit in the volcano barnacle Tetraclita rubescens. The distribution of Chthamalus in the TPE should also be introduced, before going to examples in Chile.
What are the cause the two biogeographical zones on Chile coast? Currents to create barrier or other factors? This should be introduced in the introduction.

Factors affecting the phylogeography and biogeography of intertidal species in the Pacific coast of N and S America: Along the Pacific coast of N. and S. America, there are several famous breaks, including the central America gap which is a long sandy shores that limit settlement of rocky intertidal species to pass through. There are Eastern Pacific Warm Pool and Equatorial tongue – which are oceanographic factor to limit N and S larval dispersal. This should be introduce in introduction.

What are the cause of species differentiate at 42o latitude? Oceanographic currents or geological? Should explain in the introduction.

Figures.
Fig. 1 – the figure is not clear. It should have a bigger maps as insert to show where is the Chilean coastline. The Present map also lacks scale bars and where is the direction of north. What are bold circles and thin circles as in legend? I found there are blue circle, bold circle and thin circle, what does these 3 types refer to?

Fig. 2. Can this put at the side of the map and corresponds to the map latitude, so that it can indicate which latitude is the break happens.

Discussion.
Should mention barnacles on Pacific coast of USA and TPE for comparions. E.g. Chthamalus panamensis was believed to have wide distribution. After molecular analysis, there are two distinct N and S clade, the S clade is named for a new species, C. newmani (see Chan et al. 2016, PLoS ONE).

In the discussion, last paragraph, the authors stated Notochthamalus has longer larval stage and metamorphosis than Jehlius. How much longer? Can you state the larval period for two species for information,

---

## Round 0.2 · Minor Revisions

I have gone over your revised manuscript, and find it to be generally very well revised. I have added only a few minor comments on the attached pdf, and look forward to seeing a revised version in the near future.

Reviewer 2 ·

Basic reporting

NIL

Experimental design

NIL

Validity of the findings

NIL

---

## Round 0.3 · accepted · Accept

Thank you for your revisions; this manuscript is now ready for publication. I look forward to seeing the published paper online!